# Tunable hybrid zeolites prepared by partial interconversion

Monica J. Mendoza-Castro [1], Zhipeng Qie[2,3], Xiaolei Fan [2,4,5], Noemi Linares [1] ✉ & Javier García-Martínez[1] ✉

Zeolite interconversion is a widely used strategy due to its unique advantages in the synthesis of some zeolites. By using a long-chain quaternary amine as both a structure-directing agent and porogen, we have produced superior catalysts, which we named Hybrid Zeolites, as their structures are made of building units of different zeolite types. The properties of these materials can be conveniently tuned, and their catalytic performance can be optimized simply by stopping the interconversion at different times. For cracking the 1,3,5-triisopropylbenzene, Hybrid Zeolites made of FAU and MFI units show a 5-fold increase in selectivity towards the desired product, that is, 1,3-diisopropylbenzene, compared to the commercial FAU, and a 7-fold increase in conversion at constant selectivity compared to MFI zeolite.

In some important processes, combining different zeolites enhances performance. For example, ZSM-5 is added to FCC catalysts, which contain zeolite Y, to boost its selectivity toward light olefins[1]. Similarly, the use of composite zeolite catalysts yields enhanced catalytic performance over individual zeolites in processes such as the conversion of ethanol to propylene[2], the Methanol-to-Olefins (MTO) reaction[3], the preparation of dimethyl ether from methanol[4], and the direct conversion of syngas into methyl acetate[5], among others. All these examples are based on the physical mixture of various zeolites; however, one could think of making new materials comprised of building units from multiple zeolites to combine desirable properties in one single material. Its performance could be optimized for a targeted process by tuning the percentage of each zeolite present in this solid.

There are some examples of composite catalysts containing more than one zeolite, mainly featuring core–shell structures[6–9]. In most cases, these materials are an intimate physical mixture of different zeolitic phases rather than being a single hybrid material. Zeolitic intergrowths, on the other hand, can fall under this hybrid category; however, only certain zeolites with the same building units but different layer-stacking sequences can form these materials, i.e. ERI/OFF, MFI/MEL, FAU/EMT, and others[10–12].

Ideally, one could think of making solids comprised of building units from different zeolites, which could be described as hybrid zeolites. To realize this possibility, we propose the use of the well-known interzeolite transformation method because during this process building units from both the original and the final zeolite coexist[13–15]. This does not occur in all cases. In some instances, the intermediates of these transformations are physical mixtures of the remaining parent zeolite and the new phase (both visible by XRD)[16]. However, the great variety of methods and conditions reported for interzeolite transformations, which in many instances involve the concurrent presence of building units from various zeolites, represents a simple, convenient and untapped opportunity for the production of novel, truly hybrid, and tunable catalysts[13].

It has been reported that, depending on the synthesis conditions, structural fragments of the original zeolite remain during interzeolite transformations[17]. Because of that, one can think of the use of this approach to prepare a truly hybrid material, in which building units from both the original and the daughter zeolites coexist. The evolving nature of interzeolite transformation allows the fine-tuning of the properties of these materials by simply interrupting the transformation at the suitable times. These hybrid materials should necessarily be

[1]Laboratorio de Nanotecnología Molecular, Departamento de Química Inorgánica, Universidad de Alicante, Ctra. San Vicente-Alicante s/n, 03690 Alicante, Spain. [2]Department of Chemical Engineering, School of Engineering, The University of Manchester, Oxford Road, Manchester M13 9PL, UK. [3]Faculty of Environment and Life, Beijing University of Technology, 100124 Beijing, China. [4]Nottingham Ningbo China Beacons of Excellence Research and Innovation Institute, University of Nottingham Ningbo China, 211 Xingguang Road, 315100 Ningbo, China. [5]Institute of Wenzhou, Zhejiang University, 325006 Wenzhou, China. ✉e-mail: noemi.linares@ua.es; j.garcia@ua.es

mostly amorphous, and thus no true zeolites, but rather made of building units from both the parent and final zeolites, hence the term hybrid zeolites[18].

We managed to realize the preparation of this novel family of materials by using a molecule that contains in its structure both the SDA function (tripropylammonium as a polar head) and the surfactant activity (hexadecyl as a long aliphatic chain). More specifically, we used cetyltripropylammonium bromide to drive the interconversion of FAU into MFI. The intermediates of this transformation are hybrid zeolites with excellent accessibility, provided by a large amount of well-defined and tuneable mesoporosity, and building units of both zeolites, FAU and MFI, as revealed by UV-Raman spectroscopy. These materials are not physical mixtures of both zeolites as evidenced by comprehensive characterization and by their superior catalytic performance, as compared to both pristine FAU and MFI zeolites, and physical mixtures of them.

## Results and discussion

A commercial FAU zeolite with a Si/Al ratio of ca. 40 (CBV780, the parent zeolite) was hydrothermally treated at 150 °C in an alkaline solution containing the bifunctional quaternary amine (cetyltripropylammonium bromide, CTPABr, see the "Methods" section for synthetic details and Supplementary Fig. 2 for its characterization). Samples were labeled as HyZ-*x*, where *x* indicates the time of treatment

in hours, see the "Methods" section for experimental details. The synthesized materials collected at different times were characterized to monitor the evolution of the physicochemical properties of the intermediate materials. The properties of all samples are presented in Supplementary Table 1.

The development of the MFI phase was followed by powder X-ray diffraction analysis (XRD, Fig. 1a). At short treatment times, an amorphous material was obtained (HyZ-39), which indicates that the FAU zeolite loses its long-range crystallinity during the initial stages of the transformation. As hydrothermal treatment progresses (>48 h) small peaks, due to the transformation into MFI zeolite, start to develop. Approximately, 50% crystallinity is reached after 72 h of treatment (Fig. 1d). The presence of a hump in the baseline of the intermediates suggests the presence of amorphous material. At longer crystallization times, the MFI zeolite (HyZ-96) fully forms, and the hump disappears.

**Mesoporous intermediates**

Figure 1b shows the N₂ physisorption isotherms at 77 K of the HyZ samples, whereas Supplementary Fig. 3 displays the Ar at physisorption isotherms 77 K of the same samples at logarithmic scale to better show the adsorption phenomena at low $P/P_0$. The intermediate materials prepared at 39, 48, and 72 h of transformation display type I + IV isotherms with a sharp uptake at a relative pressure of ca. 0.3, indicating the presence of both microporosity and a narrow

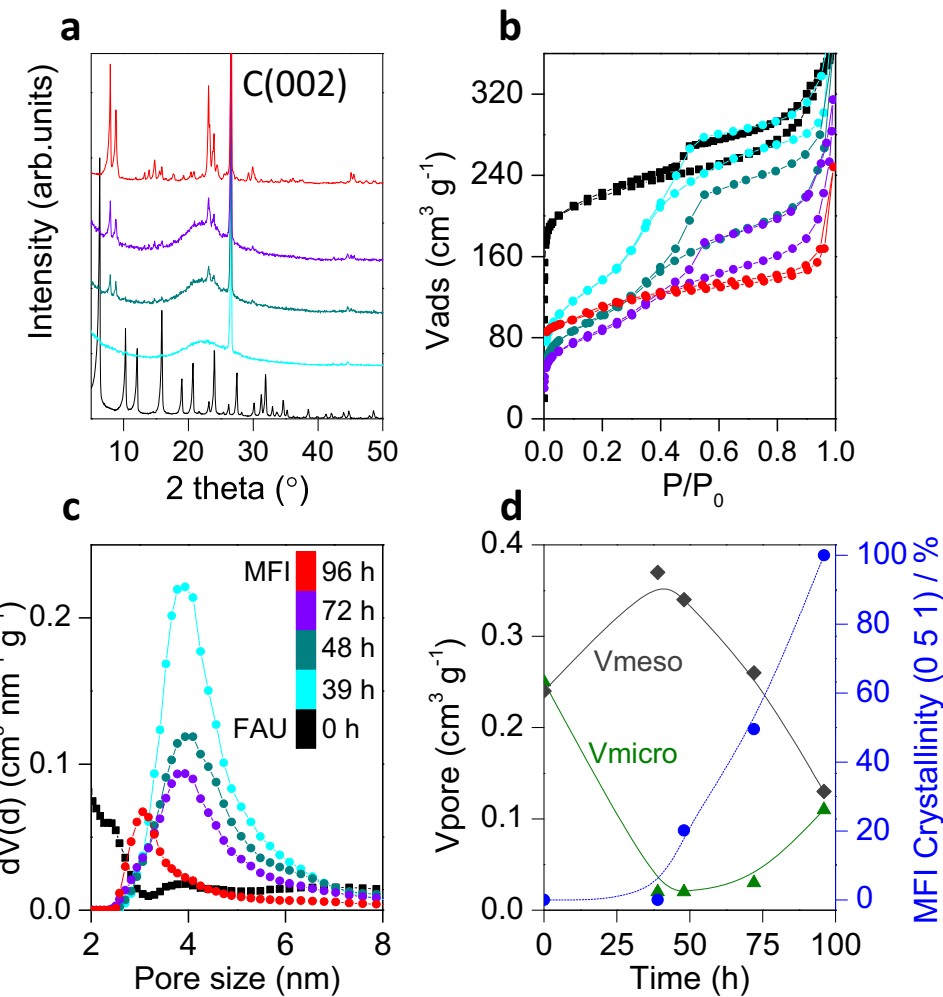

Fig. 1 | **Crystallinity and textural evolution. a** XRD patterns of solids obtained at different times during the interzeolite transformation of FAU zeolite into MFI using CTPABr as both SDA and porogen (see treatment time in **c**). The strong band at ca. 26° 2θ corresponds to the graphite used as internal standard. **b** N₂ physisorption

isotherms at 77 K for the same samples. **c** The corresponding pore size distribution calculated by NL-DFT from the isotherms. **d** Evolution of the micropore volume (green triangles), mesopore volume (gray diamonds), and MFI crystallinity (blue circles) of the samples with time.

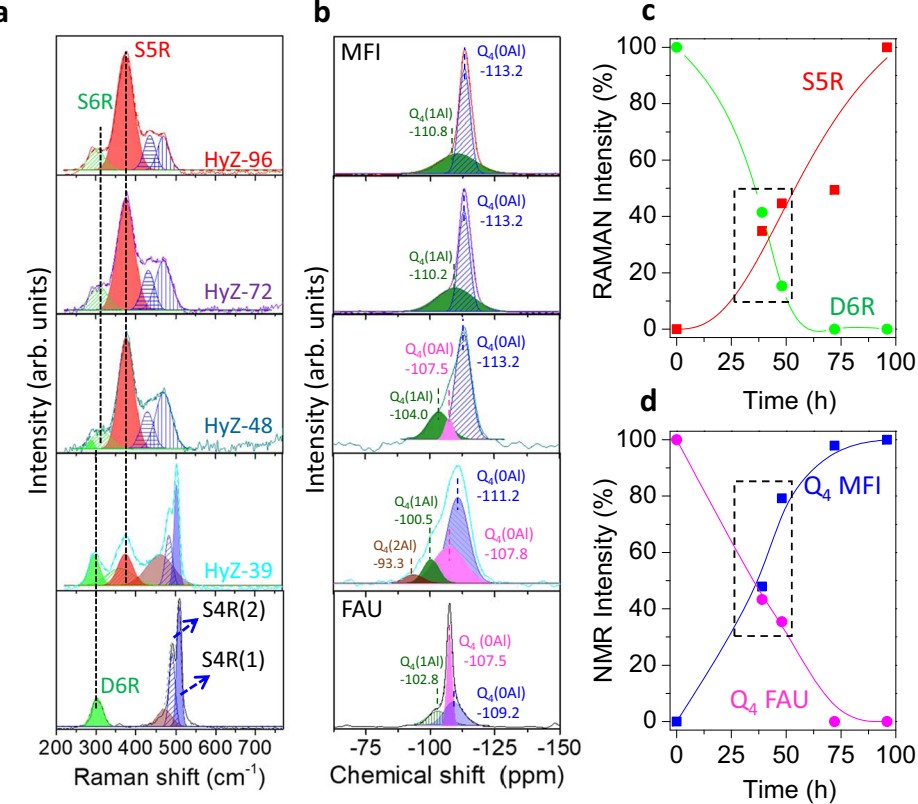

**Fig. 2 | Studying microstructural changes by UV-Raman and $^{29}$Si NMR. a** UV-Raman and **b** $^{29}$Si NMR spectra of the hybrid zeolites. **c** Evolution of the percentage of UV-Raman intensity (calculated against the intensity of the 100% crystalline materials, parent FAU and HyZ-96 for MFI) of selected bands: FAU: 300 cm$^{-1}$ for D6R, green line; and MFI: 375 cm$^{-1}$ for S5R, blue line. **d** Evolution of the percentage of Q4 intensity in the $^{29}$Si NMR spectra of the FAU (pink line) and MFI (blue line) zeolite.

distribution of mesopores centered at ~4 nm. This well-defined mesoporosity is due to the use of the bifunctional CTPABr surfactant (Fig. 1c). The textural properties of the hybrid zeolites are completely different from that of the parent FAU zeolite and the fully transformed MFI (i.e., the HyZ-96 sample). The parent zeolite displays a type I isotherm, typical of microporous materials, with an increasing uptake of N$_2$ from $P/P_0 > 0.8$, and a hysteresis loop due to the large porosity generated by the supplier during the ultrastabilization of the FAU zeolite by steaming[19]. After 96 h of treatment, a type I isotherm was obtained, revealing the formation of a new zeolite, namely MFI. The mesopore and micropore volumes of the materials follow opposite trends (Fig. 1d). While the intermediate samples present large mesopore volumes and almost no microporosity, the formation of the MFI zeolite leads to the loss of the mesoporosity and the development of microporosity. In order to study the reduction of mesoporosity with the MFI crystallization, we have followed the amount of CTPA$^+$ surfactant in the samples with the time of treatment by TGA (see Supplementary Fig. 4). The data shows that as MFI forms, the organic content of the zeolite decreases, which is further associated with the reduction of mesoporosity. There are two possible answers to this reduction. First, the increasing in the crystallinity of the sample, and the consequent densification of the network, causes micelles dissociation to supply monomers into the MFI structure, as some organics can still be found in the final MFI material. On the other hand, it has been previously reported that alkylammonium cations can undergo Hoffman degradation reactions in aqueous media at high pH values and temperatures[20] so, it is also plausible that some of the CTPA$^+$ molecules decompose during MFI crystallization following this path. It is worth noting though that in the intermediate samples, the H5-type hysteresis loops in their adsorption isotherms (Fig. 1b) indicate the

presence of mesopores accessible only through micropores, which implies that both microporosity and mesoporosity are in the same phase and not in a physical mixture.

We also followed the formation of hybrid zeolites by monitoring their microporosity at very low relative pressures (from $P/P_0 = 10^{-7}$) (Supplementary Fig. 3). FAU and MFI zeolites present very distinct adsorption profiles in the low $P/P_0$ range ($<10^{-4}$), as expected from their quite different microporous architecture. Interestingly, as a result of the interzeolite transformation process, the adsorption capacity of the intermediates at low relative pressures evolves from that characteristic of an FAU zeolite to that of an MFI zeolite, confirming the evolution of the porous architecture of the solids during treatment. The solids were studied, at different stages, using three techniques, from the more local to the one which requires longer-range order: (i) UV-Raman spectroscopy, to track the formation of the first MFI building units, as discussed later, then by (ii) XRD, which provides information about the assembly of the unit cells; and finally, by (iii) gas adsorption, as the formation of micropores requires a higher degree of order. Both, the XRD and N$_2$ physisorption data confirm the formation of amorphous mesoporous intermediates from which the MFI zeolite develops. To assess the role of the CTPABr surfactant both, in the formation of the MFI structure and the mesoporosity, experiments replacing the CTPABr by (i) CTAB, (ii) a physical mixture of TPABr and CTAB, and (iii) only TPABr, were performed (see Section 1.3 and Supplementary Table 1 at the ESI for experimental details). When only CTAB was used, no MFI was obtained even after 10 days of treatment, as shown in Supplementary Fig. 5, confirming that MFI cannot be obtained using CTAB as SDA under these conditions, as propyl instead of methyl units are needed to drive the formation of this structure. When TPABr was employed instead of CTPABr, the well-known FAU to

MFI interzeolite transformation was observed. However, this process did not evolve through mesoporous intermediates, as shown in Supplementary Fig. 6. Instead, an amorphous intermediate with very low porosity was obtained (Supplementary Fig. 6C). After 48 h of treatment, a well-formed MFI zeolite was produced. Finally, while the use of a TPABr/CTAB mixture may appear promising for simultaneous formation of MFI and mesoporosity, our results indicate that it is not sufficient for complete transformation of FAU into MFI. As demonstrated by XRD analysis and the low development of microporosity after 7 days of treatment (Supplementary Fig. 7D), only 70% of MFI was obtained. To further understand the differences between the two methods, we compared the evolution of the Si/Al ratio of the materials (Supplementary Fig. 8). The high levels of quaternary amines in the CTAB/TPABr mixture were found to inhibit the desilication of the zeolite, indicating that the initial stages of both transformations likely proceed through different mechanisms. Unlike the CTPABr case, the small change in the Si/Al ratio of the solid suggests a solid-to-solid transformation. Furthermore, TEM analysis revealed that this method produced a physical mixture of both phases: an amorphous mesoporous solid (responsible for type IV isotherms) and separate MFI crystals (Supplementary Fig. 7E). Previous research has shown that mixing conventional cationic surfactants (CTAB) with organic structure directing agents in zeolite synthesis often leads to phase separation of amorphous mesoporous material and crystalline microporous zeolite[21,22], which is consistent with our findings. Based on these observations, we conclude that only the use of a molecule that includes both a polar head (SDA group) and a mesopore-forming function (aliphatic chain)−CTPABr−allows for the interconversion of FAU into MFI and the formation of intermediate materials with well-developed mesoporosity and building units from both zeolites under equivalent synthetic conditions.

## Hybrid nature confirmation

To gain further insights into the structural properties of these materials and their evolution, UV-Raman spectra of the prepared solids were recorded, deconvoluted, and analyzed. As shown in Fig. 2a, the parent FAU zeolite (dotted black line) and the MFI daughter zeolite (red line) show the main UV-Raman bands associated with their structural building units, see schematic representation in Fig. 3a. This is, for the FAU structure: the bending mode of the double 6-membered rings (D6R) and the breathing mode of the single 4-membered rings (S4R), being (1) the S4R itself and (2) the S4R present in the D6R. The MFI sample (HyZ-96) shows, on the other hand, a sharp, high-intensity band due to 5-membered rings (S5R) units, and two low-intensity bands associated with the S4R and S6R. The deconvolution of these spectra allowed us to gain a better understanding of the changes produced. In detail, the broad peak at 470 cm$^{-1}$ in the FAU spectrum was attributed to the presence of distorted S4R units[23], which is caused by the large amount of defects present in the CBV780 structure[24]. By deconvoluting the spectrum of the MFI zeolite (HyZ-96), we identified two components in the S4R signal (434 and 466 cm$^{-1}$), which are associated to the $\nu$(Si−O−Si) and $\delta$(Si−O) vibrations, respectively. These bands have been previously identified in large MFI crystals[25], which are similar to our HyZ-96 sample that are ca. 3 µm in size MFI (see Fig. 3F). The spectra of the hybrid zeolites show a combination of these bands.

It is worth noting that the HyZ-39 sample, despite being XRD amorphous (Fig. 1a), contains building units from both zeolites, as was clearly observed by UV Raman (Fig. 2a, light blue). Three groups of bands were identified: (i) a sharp band at 503 cm$^{-1}$ with a well-defined shoulder, typical of FAU zeolites, which indicates the presence of remnant (S4R) FAU building units; (ii) a broad band at 377 cm$^{-1}$, caused by the vibration of the newly created S5R units of the MFI structure. This band is slightly shifted from the position at which it appears in highly crystalline MFI. This observation was attributed to the fact that

these S5R units are present in amorphous material, as reported elsewhere[26]. Finally, (iii) a broad band at 294 cm$^{-1}$, which is a combination of the peak due to the D6R units of the FAU structure (300 cm$^{-1}$) and the S6R units of the MFI structure (291 cm$^{-1}$) indicates the decomposition of the D6R structure into S6R units instead of S4R units[27]. This evolution was also confirmed by the decrease in the intensity of the S4R(1) band, which is due to the vibration of the S4R present in the D6R units, as aforementioned. The semi-quantification of the percentages of D6R and S5R units in the hybrid zeolites was determined by the UV-Raman band intensities at 300 cm$^{-1}$ (D6R) and 377 cm$^{-1}$ (S5R), using as references the most crystalline samples, i.e. the parent FAU zeolite and the MFI obtained at 96 h, respectively (see Fig. 2c). By doing so, we were able to confirm the coexistence of building units of both zeolites in these mesoporous hybrid zeolites. The fading of the D6R units, present in the FAU zeolite, and the development of S5R units of the MFI resulted in hybrid materials, HyZ-39 and HyZ-48, which contain different amounts of building units of both zeolites (as indicated by the squared region in Fig. 2c).

The same conclusion can be drawn from $^{29}$Si NMR analysis (Fig. 2b), which corroborates the hybrid nature of samples HyZ-39 and HyZ-48 (Fig. 2d). The two main resonances found in the pristine zeolites were associated to Q4(0Al), in agreement with their low Al content (Si/Al ca. 40); however, the chemical shifts for both structures are rather different (i.e., −107 ppm for FAU and −113 ppm for MFI, respectively)[24,28–30] which we used to monitor the formation of the MFI zeolite. At short times of treatment, (i.e., the sample HyZ-39), the characteristic band of Q4(0Al) in FAU becomes broader and less intense, as a consequence of the loss of long-range order in this material, which agrees well with our XRD analysis (Fig. 1a). At the same time, the small signal due to less ordered Q4(0Al) (at −109 ppm in FAU) develops and shifts (−111 ppm in HyZ-39) to values near to the ones found for the MFI structure. Both observations confirm the hybrid character of this sample containing building units of the original and the newly formed zeolites. As the treatment time increases, the Q4(0Al) signal shifts to the MFI position (−113 ppm from HyZ-48), whilst the FAU signal gradually fades out. A smaller Q4(1Al) resonance was observed in all the solids, displaying a similar trend. It is worth pointing out that the sample with the lowest Si/Al ratio (HyZ-39) also presents a very small band due to Q4(2Al) as a result of the partial desilication that occurs at short times of treatment.

When physical mixtures of FAU and MFI zeolites were characterized by Ar physisorption at 77 K, and both UV-Raman and $^{29}$Si NMR spectroscopies, completely different results were obtained (Supplementary Fig. 9). The physical mixtures, prepared at different FAU/MFI ratios, present a bimodal microporosity, as expected of a mixture of materials with different micropore sizes. Similarly, the physical blends yield both UV-Raman and $^{29}$Si NMR spectra that reveal a linear combination of the bands of building units present in both zeolites and not a continuous evolution of intermediates which further confirms the hybrid (no blended) nature of our materials.

## Mechanistic insights

In addition to the structural and textural evolution of the samples, their morphology is also markedly altered during the interzeolite transformation. Figure 3 shows a schematic representation of the proposed mechanism of this process and the morphological evolution of the samples by SEM, TEM, and STEM-EDX mapping analysis. As previously reported[16], one of the mechanisms of interzeolite transformation involves the partial dissolution of the FAU zeolite (Fig. 3b) to form amorphous materials, as evidenced by Fig. 3c, even if the morphology of the FAU crystal is maintained in some cases. The analysis of the mother liquor at different treatment times (see Supplementary Fig. 8) confirms the partial desilication of the parent FAU zeolite at short times of treatment, so we concluded that the initial steps of transformation, in this case, proceed via a solution-mediated

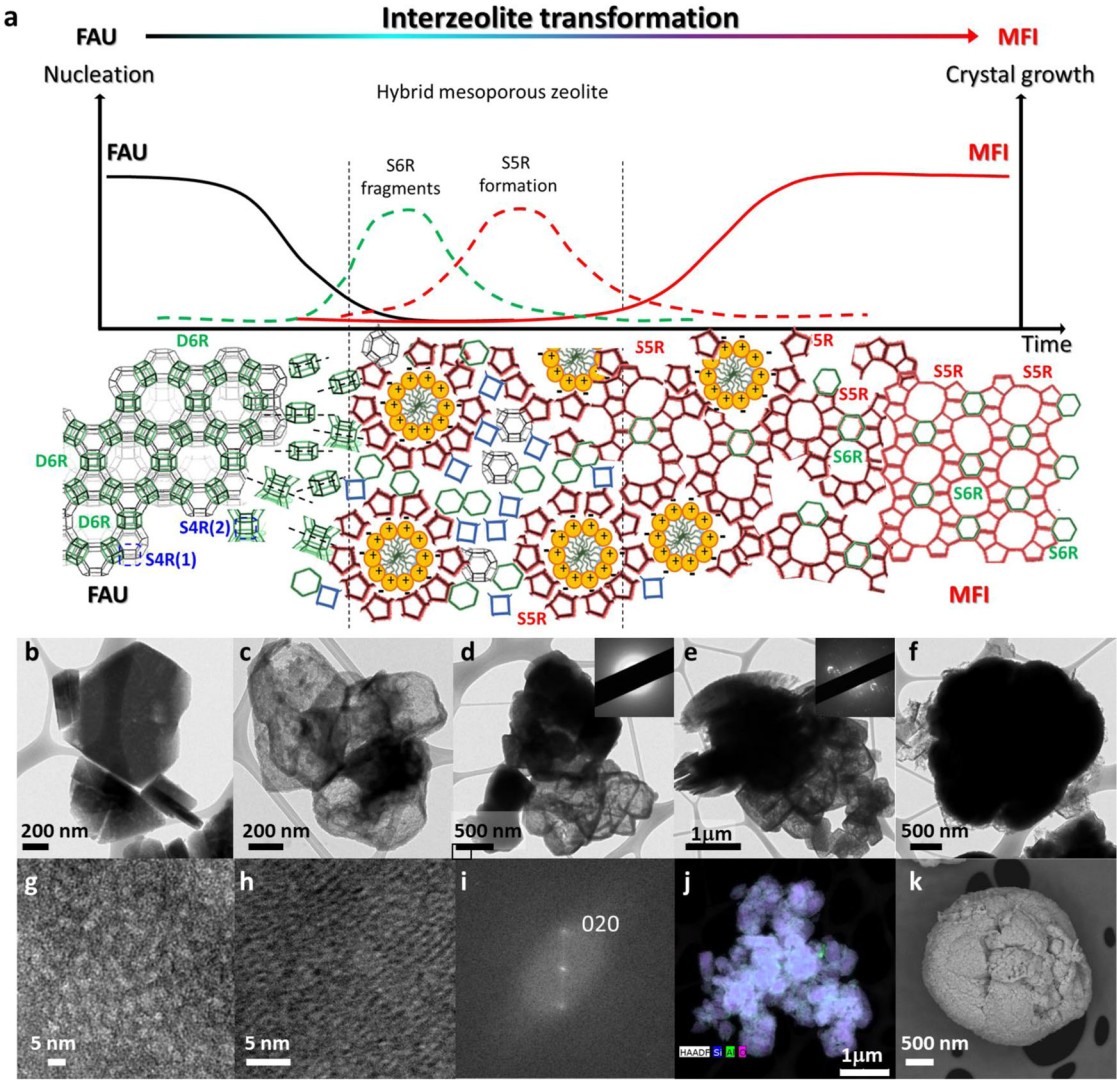

**Fig. 3 | Revealing the morphology of hybrids and a plausible mechanism of formation. a** Schematic representation of the various stages of the FAU transformation into MFI zeolite directed by the CTPABr. **b**–**f** TEM micrographs of samples obtained at different times of treatment, from left to right, 0, 39, 48, 72 and 96 h. **g** and **h** Higher magnification TEM micrographs of ultramicrotomed samples prepared 48 and 96 h of treatment. **i** The FFT corresponding to image **h** showing spots ascribed to the (020) plane of the MFI zeolite structure. **j** STEM-EDX mapping of sample HyZ-72 showing the homogeneous distribution of Si (blue) and Al (green). **k** FE-SEM micrograph of sample HyZ-96.

mechanism[13]. From the dissolved fragments the new zeolite can be formed. In fact, as reported elsewhere[31], the formation of S5R units from the FAU zeolite, which does not contain such structural units, should occur via dissolved silicates coming from a medium or high-silica zeolite. The development of MFI peaks in the XRD patterns occurs as a denser phase forms from the XRD amorphous solids that maintain the FAU morphology (HyZ-48). This dense phase is initially amorphous, as revealed by SAED, Fig. 3d, but evolves to form MFI crystals, first relatively small (HyZ-72, Fig. 3e) and subsequently significantly larger and agglomerated (HyZ-96, Fig. 3f). Due to the presence of CTPABr, mesoporosity develops while the MFI zeolite emerges, see a high magnification micrograph of HyZ-48 in Fig. 3g. Finally, large (2−3 μm) crystals were obtained, as shown in Fig. 3f and k. As abovementioned, the final material, while being totally crystalline, lacks mesoporosity, see the crystalline structure of the final sample in Fig. 3h and i. In the final stage of the transformation, the crystallization occurs by a solid−solid mechanism (as evidenced by the high solid yield in all cases, >80%, see Supplementary Table 1, and the maintaining and homogeneity of the Si/Al ratio, Figs. 3j and S8). It is well known that zeolites crystallize from amorphous precursors via a

nonclassical mechanism[32], which involves the intimate contact of the various phases, so the assembly and attachment of building units or particles from the amorphous solid to the new MFI zeolite cannot be discarded (see Supplementary Figs. 10 and 11).

## Tunable catalytic enhancement

To assess the potential of the hybrid zeolites to convert bulky molecules, they were tested for the catalytic cracking of 1,3,5-triisopropylbenzene (TiPBz) (Fig. 4). This is a model reaction to evaluate the accessibility of zeolites, as TiPBz (kinetic diameter = 0.94 nm) cannot access their narrow microporosity[33]. As control, the parent FAU zeolite, namely CBV780, was also included. This USY zeolite is highly mesoporous and typically yields excellent catalytic cracking results[1]. As shown in Fig. 4a, the hybrid HyZ-48 sample presented even higher conversion than the CBV780, a benchmark catalyst for this reaction. This optimized catalyst (HyZ-48) features a combination of high mesoporosity and building units of both zeolites, which enhances the conversion over materials either more crystalline−but with little or no mesoporosity (HyZ-72)−or with higher mesoporous volume−but very poor or no crystallinity (HyZ-39). Moreover, all the hybrid zeolites

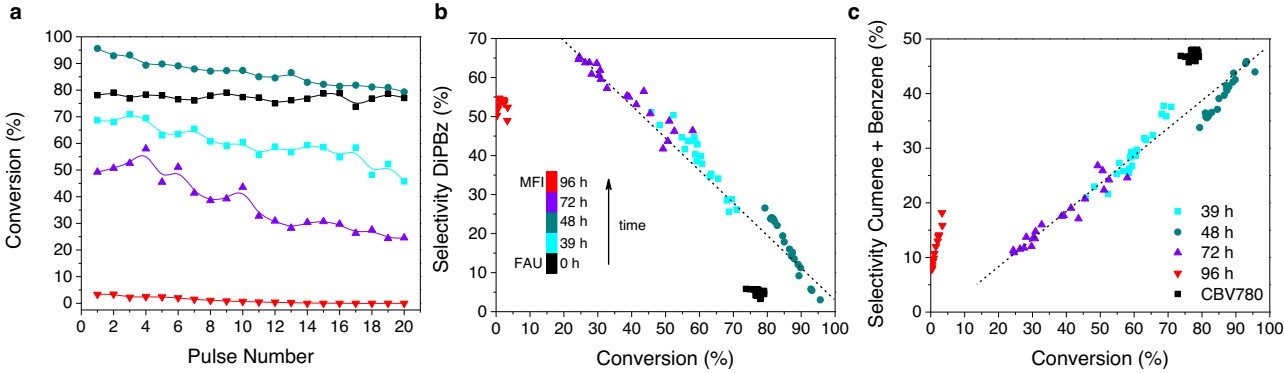

**Fig. 4 | Tunable catalytic activity of the hybrids. a** Performance of the materials in the catalytic cracking of TiPBz. **b** Evolution of the selectivity for 1,3-diisopropylbenzene with the increased conversion of TiPBz. **c** Evolution of the selectivity for cumene and benzene with the increased conversion of TiPBz.

exhibit significantly higher conversions than the MFI zeolite (HyZ-96), owing to their higher accessibility. More importantly, hybrid zeolites also yield improved selectivity than conventional zeolites. FAU and MFI produce lower amounts of DiPBz (the target compound) at the same TiPBz conversion (solid black squares and red inverted triangles in Fig. 4b). For example, CBV780 (black squares) yields ca. 5% of DiPBz at 80% TiPBz conversion. Conversely, at a similar conversion, our HyZ-48 catalyst is five-fold more selective towards the desired product, ca. 25% (dark cyan circles). The same trend is true for the MFI zeolite. More specifically, 50% DiPBz was obtained when using these catalysts, but only at a very low conversion (<5%), while HyZ-72 yields the same selectivity at a much higher (45%) conversion. The amount of DiPBz produced after 20 pulses by the intermediates further evidence the superior catalytic performance of these solids. While HyZ-39, HyZ-48 and HyZ-72 yield, respectively, 158, 95 and 140 mmol of DiPBz, the original CBV780 results in only 26 mmol and the daughter MFI (HyZ-96) affords the worst performance with just 3.9 mmol of DiPBz obtained. The longer diffusion path lengths of the conventional zeolites (both FAU and MFI) cause the overcracking of the reaction intermediates reducing the formation of the desired product, DiPBZ, which is used as a proxy of gasoline and diesel in the cracking of vacuum gas oil[33]. Regarding the selectivity of undesired compounds (cumene and benzene), an opposite trend was observed. Hybrid zeolites always yield lower amounts of these two unwanted overcracked products than FAU and MFI zeolites at the same conversion, as shown in Fig. 4c. Figure 4b and c also illustrate that our method allows for modulating the physicochemical characteristics of hybrid zeolites and by doing so fine-tuning their catalytic performance.

For comparison purposes, physical mixtures of FAU and MFI and a mesoporous aluminosilicate with a similar Si/Al ratio (ca. 40) were also tested for the cracking of TiPBz (Supplementary Fig. 12). The conversion levels of the physical mixtures and the intermediate materials are similar. As expected, the higher the amount of FAU in the mixture, the higher the conversion obtained, owing to its larger micropore size (~0.74 nm) as compared to MFI (~0.54 nm). However, the selectivities of the physical mixtures of conventional zeolites fall in the line between both zeolites indicating that, for the same conversion, lower selectivities are always obtained for the physical mixtures as compared to our catalysts, which confirms the hybrid nature of the intermediates. On the other hand, a typical amorphous mesoporous aluminosilicate (Al-MCM-41) yields similar selectivities than the intermediates but very low conversion when compared to both the zeolites and the intermediate. From these results we conclude that our materials present the best of two worlds, this is, similar conversions to highly acidic zeolites and excellent selectivities to DiPBz which is a feature of highly accessible mesoporous materials.

These results show that hybrid zeolites are highly active and selective catalysts for the conversion of bulky molecules, which further

evidences their enhanced accessibility and strong acidity. To confirm the zeolite-like acidity in our solids, $^{27}$Al NMR analyses were performed to study the different Al environments on the materials (Supplementary Fig. 13). The $^{27}$Al NMR spectrum of the original FAU sample shows the usual two bands around 60 and 0 ppm due to the presence of Al in tetrahedral (Al(IV)) and octahedral (Al(VI)) coordination environments, respectively. Noticeably, the samples prepared by interzeolite transformation do not contain octahedral Al (i.e., extra-framework species). The transformation of octahedral Al into tetrahedral Al has been widely reported for the treatment of zeolites with alkaline solutions and associated to the incorporation of extra-framework Al into the zeolite framework[34–36]. The broadening of Al(IV) signal is due to the formation of partially bonded framework Al, that is, with higher disorder, which was also found in the $^{29}$Si NMR analysis (Fig. 2b). Interestingly, the $^{27}$Al NMR signal shifts to values more typical of the MFI structure, which indicates that the local environment of the Al sites evolves towards that found in an MFI structure. This confirms the Raman spectroscopy results (Fig. 2a), even at short times (39 h), when XRD peaks are not observed. Finally, Al-MCM-41 material contains ca. 50% of octahedral aluminum (Al(VI)), while even our XRD amorphous material (HyZ-39) only contains tetrahedral Al(IV). The exclusive presence of only Al(IV) in our intermediate materials can be related to their superior catalytic performance, as suggested by the much higher conversion of HyZ-39 vs. Al-MCM-41 (60% vs. 10% after 10 pulses, Supplementary Fig. 13) both showing similar porosity and lacking XRD crystallinity.

In conclusion, a novel family of superior catalysts (hybrid zeolites) containing building units of structurally very different zeolites were prepared by partial interzeolite transformation. The use of CTPABr allowed for both the coexistence of building units of FAU and MFI zeolites and the development of well-defined mesoporosity. Control experiments with only CTAB or TPABr failed to produce the complete interconversion of FAU into MFI, while a mixture of them produced a physical mixture of the MFI zeolite and a mesoporous amorphous material, which confirms the need to have the SDA group (polar head) and the surfactant function (aliphatic chain) in one single molecule to produce true hybrid zeolites. These materials displayed superior catalytic performance for the cracking of TiPBz. The combination of well-defined mesoporosity and the presence of zeolitic building units in their structure resulted in increased production of desired products (DiPBz) and in a reduction of the unwanted ones (cumene and benzene). The observed reduction in overcraking reactions is likely due to the ready exit of the reaction intermediates, in this case, DiPBz, from our catalysts, as observed in other hierarchical solids. The possibility of tuning the physicochemical properties of the hybrid zeolites by kinetic control of the interzeolite transformation opens up new opportunities for the development of highly active and selective catalysts for the conversion of bulky molecules. This paves the way for the fabrication

of hybrid hierarchical catalysts with optimized properties for those processes in which the combined use of different zeolites yields improved performance.

## Methods

### Materials

For the preparation of the mesoporous materials, USY Zeolite (CBV780) was supplied by Zeolyst, with a nominal Si/Al ratio of 40, and tetrapropylammonium bromide 98% was purchased from Sigma-Aldrich (St. Louis, MO). Sodium hydroxide (98% pellets) was supplied by Fluka.

For the synthesis of the quaternary ammonium bromide surfactant with the tripropyl head, tri-n-propylamine and 1-bromohexadecane both from Sigma-Aldrich were employed.

For the catalytic evaluation, 1,3,5-triisopropylbenzene ($C_{15}H_{24}$, 95%, Sigma-Aldrich) was used as feed stock. The chemicals used for GC calibration are:

From Sigma-Aldrich: Benzene ($C_6H_6$, ≥ 99.8%), toluene ($C_6H_5CH_3$, ≥ 99.5%), para-xylene ($C_6H_4(CH_3)_2$, ≥99.5% GC), ortho-xylene ($C_6H_4(CH_3)_2$, ≥99.5% GC), meta-xylene ($C_6H_4(CH_3)_2$, ≥99.5% GC), 1,2,3-trimethylbenzene ($C_6H_3(CH_3)_3$, ≥99.5%, neat, GC), 1,2,4-trimethylbenzene ($C_6H_3(CH_3)_3$, 98%) and, 1,3-diisopropylbenzene ($C_{12}H_{18}$, 96%).

From Alfa Aesar: cumene ($C_9H_{12}$, 99%) and, 1,4-diisopropylbenzene ($C_{12}H_{18}$, 99%).

### Synthesis of the cetyltripropylammonium bromide surfactant

For the synthesis of this quaternary ammonium bromide surfactant, tri-n-propylamine was reacted with 1-bromohexadecane as described elsewhere[37]. Briefly, tripropylamine (1 equiv.) and n-Hexadecyl bromide (1.2 equiv.) were refluxed in a solvent mixture of 30% methanol/acetonitrile for 48 h. Then the solvents were removed on a rotary evaporator, and the mixture was crystallized from methanol/diethyl ether (2:10) three times to get the pure product.

### Interzeolite FAU to MFI transformations

Different methods were evaluated for the transformation of FAU zeolite into MFI in order to analyze the physicochemical properties of the intermediates. First, the use of the cetyltripropylammonium bromide as both surfactant and SDA was explored. In a typical synthesis, 0.374 g of CTPABr was dissolved in 28.5 mL of an aqueous NaOH solution (0.08 M). Once the surfactant was totally dissolved, 1 g of CVB780 (FAU) zeolite was added to the mixture and stirred for 1 h at room temperature. The mixture was transferred to a Teflon-lined stainless-steel autoclave and statically heated at 150 °C for 39–96 h.

In a second approach, a mixture of the surfactant CTAB and TPABr were used for the FAU to MFI transformation. In a typical synthesis, 0.4 g of CTAB and 0.222 g of TPABr were dissolved in 28.5 mL of an aqueous NaOH solution (0.08 M). Once the organic compounds were totally dissolved, 1 g of CVB780 (FAU) zeolite was added to the mixture and stirred for 1 h at room temperature. The mixture was transferred to a Teflon-lined stainless-steel autoclave and statically heated at 150 °C for 6–168 h.

Finally, two control experiments were performed: (1) CTAB was evaluated as SDA for the transformation of FAU zeolite into MFI and (2) a typical TPABr-directed interzeolite transformation was carried out. For the first one, 0.3 g of CTAB was dissolved in 28.5 mL of an aqueous NaOH solution (0.08 M). Once the CTAB was totally dissolved, 1 g of CVB780 (FAU) zeolite was added to the mixture and stirred for 1 h at room temperature. The mixture was transferred to a Teflon-lined stainless-steel autoclave and statically heated at 140 °C for 4–8 days. For the second one, 0.222 g of TPABr was used instead of the CTAB. The rest of the procedure was equivalent, but the hydrothermal treatment was performed only up to 2 days.

In all cases, the final material was filtered, washed with distilled water, dried at 60 °C overnight and calcined at 550 °C for 5 h (2 °C min$^{-1}$).

### Materials characterization

X-ray diffraction (XRD) patterns were obtained in a powder X-ray diffractometer (Bruker AXS D8 Advance) with graphite monochromatized Cu Kα radiation at 40 kV and 40 mA. A known amount of graphite (10 wt%, used as an internal standard) was mixed with the samples before the analysis in order to quantify the crystallinity of the different samples (see the peak corresponding to the graphite at ca. 26° 2θ)[38]. The most crystalline MFI zeolite obtained after interzeolite transformation is completed, this is zeolite prepared by using CTPABr as SDA at 96 h, was defined as 100% crystallinity and used as a reference to calculate the percentage of MFI in the intermediate samples. The formula used to calculate that percentage was: $\%MFI_{(051)} = \frac{I_{(051)x}/I_{(graphite)x}}{I_{(051)ref}/I_{(graphite)ref}} *100$. Nitrogen physisorption isotherms at 77 K were performed in a Quadrasorb-Kr/MP apparatus from Quantachrome Instruments. The samples were previously degassed for 4 h at 250 °C at $5 \times 10^{-5}$ bar. Adsorption data were analyzed using the software QuadraWinTM (version 6.0) of Quantachrome Instruments. Cumulative pore volumes and pore-size distribution curves were calculated by applying the NL-DFT method to the adsorption branch of the isotherms. The total pore volume was obtained at the plateau of the cumulative adsorption pore volume plot at a relative pressure ($P/P_0$) of 0.8 to discard the large mesoporosity due to the steaming present in the original USY. Micropore volume was determined by NL-DFT as the volume adsorbed at pore sizes <2 nm and mesopore volume was calculated by subtracting the micropore volume from the total pore volume as shown in ref. [39]. Argon physisorption isotherms at 77 K were performed with high resolution in the low partial pressure region by using a 3FLEX apparatus from Micromeritics. The morphology of the samples was evaluated by transmission electron microscopy (TEM) images collected using a JEM-1400 Plus microscope (JEOL, 120 kV, 0.38 nm of resolution). STEM-EDX analyses for the study of the compositional differences were performed in an FEI Talos F200X microscope (200 kV). The UV-Raman spectra were recorded on a Jasco NRS-5100 dispersive Raman system with a laser source of 325.03 nm; exposure of 10 s with a Z position of 15956 μm and an accumulation of 100 scans. The composition of the prepared solids was calculated by subtracting the amount of Si and Al in the reaction mixtures (determined by inductively coupled plasma emission or mass spectroscopy (ICP-OES/MS)) to the Si and Al content of the parent FAU zeolite. The concentration of Si in the liquids was determined by an Optima 7400DV ICP-OES spectroscope from Perkin Elmer, while the Al content was analyzed in an Agilent 7700x ICP-MS spectroscope. MAS NMR spectra of the solids were recorded on a Bruker ADVANCE III HD 500 MHz (ν$^{29}$Si = 99.36 MHz), with a CPMAS 4 mm zirconia probe. The following parameters and pulse sequences were used: $^{29}$Si-DP: Direct one pulse polarization without decoupling. π/4-pulse pulse duration−3 μs, relaxation delay−10 s, number of scans−576. Reference: tetramethylsilane−0 ppm.

### Catalytic evaluation

Catalytic cracking of 1,3,5-triisopropylbenzene (TiPBz) over the zeolitic materials under investigation was conducted using a pulse method based on gas chromatography (GC, Varian 3400), as described elsewhere[40]. The schematic of the experimental rig used for the pulse experiments is shown in Supplementary Fig. 1a. To identify the retention time (RT) of all products, GC was calibrated first by injecting standards including benzene, toluene, o,m,p-xylene 1,2,3- and 1,2,4-trimethylbenzene (TMB), 1,3- and 1,4-diisopropylbenzene (DiPBz) and TiPBz (Supplementary Fig. 1b). Before experiments, zeolites were pelletized, carefully crashed, and sieved

to ~250 mesh (i.e., 63 µm) particle size. Then, 18 mg zeolite particles were loaded in an inlet liner (Restek 20793, i.d. = 4 mm, o.d. = 6.3 mm and length = 72 mm) made up of borosilicate glass, in which deactivated glass wool (Restek) was filled to hold the catalyst bed. The inlet liner was inserted into the GC and heated to 375 °C, keeping for 2 h to remove the inherited moisture in zeolites. In a typical pulse experiment, 0.2 µL of TiPBz was injected manually using a syringe (Thermo Fisher, 1 µL range), which was vaporized and carried to catalyst bed by 200 mL min$^{-1}$ helium (He) gas flow. Partial flow of the products went through the GC column (Stabilwax i.d. = 0.32 mm, length = 30 m, film thickness = 1 µm) under a split ratio of 200:1, and then were analyzed online by flame ionization detector (FID) at 300 °C. Specifically, the initial temperature of the GC column was 80 °C, then increased to 220 °C at a ramp rate of 10 °C min$^{-1}$, and maintained for 10 min. In total, the analysis time for one injection was 30 min, which was repeated 30 times for one sample.

## Data availability

The datasets generated during the current study are provided in the Source Data file. Source data are provided with this paper.

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

## Acknowledgements

This project has received funding from the European Union's Horizon 2020 research and innovation program under grant agreement No 872102. M.J.M. thanks the Generalitat Valenciana for a Ph.D. fellowship (GRISOLIAP/2020/165). Z.Q. thanks the financial support of the China Scholarship Council for his Ph.D. secondment at The University of Manchester (CSC file no. 201906120207). The authors thank Abdullah Alhelali at the Department of Chemical Engineering, The University of Manchester for performing the additional catalytic tests.

## Author contributions

M.J.M. synthesized the materials, as well as conducted morphological and structural characterizations, and M.J.M., N.L. and J.G.M. realized the corresponding analysis. Z.Q. performed the catalytic experiments, and Z.Q. and X.F. realized the corresponding analysis. All the authors participated in the manuscript writing. J.G.M. and X.F. obtained the funding. N.L. and J.G.M. conceived the idea and supervised the study.

## Competing interests

The authors declare no competing interests.
