## [Peer Review File · Nature Communications]

Tunable hybrid zeolites prepared by partial interconversionREVIEWER COMMENTS

Reviewer #1 (Remarks to the Author):

This paper studies the tuneable formation of hybrid mesoporous zeolites through the interzeolite transformation approach. In particular, the evolving mechanism from FAU to MFI zeolite is clearly monitored using combined techniques at different scales. I recommend publishing this manuscript after major revision. Some specific questions are as follows.

1. The caption of Figure 1D lacks crystallinity.
2. In line 157, the UV-Raman bands are shown in Figure 2A instead of Figure 2B.
3. Since the author mentioned the evolution of Q4(1Al) and Q4(2Al) bands in ²⁹Si NMR analysis, their chemical shifts should be labelled in Figure 2 or pointed out in the context.
4. To better compare the catalytic activity, the yields of target products on various zeolites could be complemented in Figure 4.
5. The legends of Figures S3, S5 and S6 are partially or totally missing.

Reviewer #2 (Remarks to the Author):

In this work, Mendoza-Castro et al. use CTPABr as a structure directing agent and porogen to obtain mesoporous hybrid zeolites with FAU and MFI structures. The crystallization mechanism was proposed and demonstrated by a series of experiments and characterizations. Although some new interesting results are obtained, some similar results have been reported in the authors' very recent work (FAU to BEA, *J. Am. Chem. Soc.* 2022, 144, 5163–5171). We think this manuscript cannot be accepted by Nature Communications (a high-level journal). Also, some questions need to be confirmed:

1. The cetyl groups (e.g. CTAB) have been widely reported to produce mesopores over the past several decades.
2. On the basis of the authors' data, the formation of tetrapropyl units might be the key factor to determine the growth of MFI zeolite from FAU zeolite. However, there is no information about the formation of tetrapropyl units. Furthermore, the mesopores are dispersed in the 96 h sample. Why? Are the cetyl groups decomposed?
3. In Figure S6, the FAU zeolite was amorphous after 6 h in CTAB+TPABr, and a relatively high crystallite MFI zeolite was formed after another 6 h (Figure S6), which are a little faster than those in CTPABr (39 and 48 h, Figure 1), indicating the mechanisms of CTAB+TPABr and CTPABr are similar. The differences might be caused by the different dissolved rates of FAU zeolites in CTAB+TPABr and CTPABr.
4. Authors claim that only the use of a molecule comprised of both, the SDA group (polar head) and a mesopore forming function (aliphatic chain), that is, CTPABr, allows for interconversion of FAU into MFI and yields intermediate materials containing well-developed mesoporosity and building units from both zeolites. However, similar samples are obtained using CTAB+TPABr (Table S1 and Figure S6).
5. The data in Table S1 (12, 24, 72, and 168 h) is not in good accordance with Figure S6 (6, 12, 36, and

168 h).

6. A strong peak at ~ 260 can be observed in all the XRD patterns. There is no information about this strong peak.
7. In Figure S3, there are six samples. However, only five colors are identified.
8. Figures S2, "CTEABr" should be "CTPABr".
9. In page 5, line 147~150, Figure S5D and S5E should be Figure S6D and S6E, respectively.
10. In Figure 2, the authors claimed that time-resolved UV-Raman spectra of the prepared solids were recorded. However, no details of the testing conditions are provided. Furthermore, these spectra are not in situ results and thereby cannot be named as time-resolved UV-Raman spectra.
11. The letters (a, b, c, ...) in all Figures should be capitalized (A, B, C, ...).

Reviewer #3 (Remarks to the Author):

The authors reported the transformation of FAU zeolite into ZSM-5 zeolites. The transformation of one type of zeolites to another is well known in the open literature. However, the authors reported the influence of different amines on the formation of ZSM-5 and the generation of porosity to the intermediates amorphous materials. Moreover, the study showed the importance of an amorphous material in which most researchers neglect their use. Therefore, this research showed some originality. The structure of the starting material, intermediates (named as hybrid materials), and final product were first prepared and then investigated using different characterization techniques. Although the structure was fairly characterized, the manuscript required MAJOR modifications before being considered to be published:

- The experimental is not well explained; if someone wants to prepare the material, I believe he/she cannot. In the "Interzeolite FAU to MFI transformations" section (p. 11, lines 343-345), the amount of each solution that constitutes the 28.5 mL solution must be stated clearly NOT only mentioning the concentration. Please do the same for other surfactants. The equation used for calculating the percent of crystallinity must be stated. Furthermore, an alternative and reliable sophisticated technique for calculating the percent of MFI and amorphous should be used instead of relying on XRD results.
- The setup used for the evaluation of the catalysts was not well explained. The authors need to draw a detail schematic for the reaction setup with products analysis.
- "Scheme 1" in the introduction was mentioned and I do not see it in the text.
- There are some typos that need to be corrected.
- In line 105, Figure S3 is Ar physisorption NOT N2.
- Lines 124-125, the statement is not in agreement with results represented in Figure S3; the intermediate materials showed a higher uptake at low relative pressure, say $P/P_0 = 1E-5$, in comparison with FAU ref.
- Figure S5 is not well labeled. It is recommended to put all figures as in Figure S5C.
- Figure 2B is for ^{29}Si NMR, line 157.
- All SEM and TEM micrographs should have the same resolution. Also, it's preferable to label the scale bar rather than mentioning that in the caption. The inter-structure of the produced materials should be

confirmed by high-resolution TEM. You can find similar work in the literature

<https://doi.org/10.1016/j.fuel.2018.03.161> .

- “These MFI building units grow over the remains of the FAU zeolite (Figure 3D)”: the authors showed the dissolved fragments of the FAU zeolites over the particles, but failed to show the MFI particles? If they claim that an x % of MFI was formed, SEM and TEM micrographs should show the morphology of both mixtures and should be compared to the physical mixing.
- “As the MFI zeolite forms, the surfactant micelles are expelled out of the crystal, which causes the final material to lack mesoporosity” how does this happen?
- The evaluation part starting from the experimental, as mentioned earlier, still needs to be improved. The authors need to talk about the active sites responsible for the catalytic cracking of TiPBz and how these active sites differ from one sample to another. In fact, the chemical properties of all catalysts were not characterized. The porosity of a material is important to access active sites and controlling the selectivity of the products. This was clear from the reaction results over ZSM-5, which showed limited activity because the feed was unable to access the active sites. The role of acidity should be highlighted, and catalysts acidity should be determined and linked to the reaction results. The physically mixed FAU and MFI should be evaluated in the reaction and compared to the HyZ materials.

REVIEWER COMMENTS

Reviewer #1 (Remarks to the Author):

This paper studies the tuneable formation of hybrid mesoporous zeolites through the interzeolite transformation approach. In particular, the evolving mechanism from FAU to MFI zeolite is clearly monitored using combined techniques at different scales. I recommend publishing this manuscript after major revision. Some specific questions are as follows.

Responses to Reviewer 1: We wish to thank the reviewer for their helpful comments and for recommending the publication of our manuscript. We have made changes to the manuscript to address the reviewers' suggestions.

Comment 1. The caption of Figure 1D lacks crystallinity.

Response 1: The caption has been expanded to include information about the crystallinity.

Comment 2. In line 157, the UV-Raman bands are shown in Figure 2A instead of Figure 2B.

Response 2: The typo has been corrected.

Comment 3. Since the author mentioned the evolution of Q4(1Al) and Q4(2Al) bands in ²⁹Si NMR analysis, their chemical shifts should be labelled in Figure 2 or pointed out in the context.

Response 3: The chemical shifts of the all bands have been identified and correctly labelled in Figure 2.

Comment 4. To better compare the catalytic activity, the yields of target products on various zeolites could be complemented in Figure 4.

Response 4: We agree with the reviewer that including the yields of target products adds to the discussion of Figure 4 and better reflects the enhancement of the catalytic performance for the intermediate catalysts. Therefore, we have calculated the amount of diisopropylbenzene (desired product) that is produced by all the catalysts and we found that the intermediate catalysts yield a 4-fold increase in the production of that compound. The following sentence has been then incorporated to the main text:

“The amount of DiPBz produced after 20 pulses by the intermediate further evidences the superior catalytic performance of these solids. While HyZ-39, HyZ-48 and HyZ-72 yield, respectively, 158, 95 and 140 mmol of DiPBz, the original CBV 780 results in only 26 mmol and the daughter MFI (HyZ-96) affords the worst performance with just 3.9 mmol of DiPBz obtained. The longer diffusion path lengths of the conventional zeolites (both FAU and MFI) cause the overcracking of the reaction intermediates reducing the formation of the desired product, DiPBZ, which is used as a proxy of gasoline and diesel in the cracking of vacuum gas oil.^[31]”

Comment 5. The legends of Figures S3, S5 and S6 are partially or totally missing.

Response 5: The legends on the figures in the supporting material have been added to the corresponding figures.

Reviewer #2 (Remarks to the Author):

In this work, Mendoza-Castro et al. use CTPABr as a structure directing agent and porogen to obtain mesoporous hybrid zeolites with FAU and MFI structures. The crystallization mechanism was proposed and demonstrated by a series of experiments and characterizations. Although some new interesting results are obtained, some similar results have been reported in the authors' very recent work (FAU to BEA, *J. Am. Chem. Soc.* 2022, 144, 5163–5171). We think this manuscript cannot be accepted by Nature Communications (a high-level journal). Also, some questions need to be confirmed:

Responses to Reviewer 2: We wish to thank the referee for their time to review this manuscript and for their constructive feedback, which have enhanced the quality of the manuscript. However, before answering the referee's comments we would like to point out the differences between our manuscript and the previous work cited by the reviewer.

In our JACS article (*J. Am. Chem. Soc.* 2022, 144, 5163–5171), we described how the partial transformation of FAU into BEA results in poorly crystalline mesoporous materials, irrespectively of the different approaches we carried out. Notably, none of these approaches involved the use of a compound which is, at the same time, the structure directing agent for the formation of the zeolite and a surfactant, as we report in this new work. In the case of the transformation of FAU into MFI, the only method that yields mesoporous catalysts featuring mesoporosity and FAU and MFI units requires the use of this bifunctional molecule. More interestingly, the FAU into BEA transformation proceeded via the complete loss of the FAU phase before the BEA phase starts to form. There was no coexistence of both phases. What makes this new work more exciting, is the fact that we have prepared intermediate materials containing fragments of both zeolites, for this reason we call them 'hybrid zeolites'. The sample prepared at 39 h of transformation, for example, shows double 6-membered rings (D6R) coming from the FAU structure, coexisting with 5-membered rings (S5R) units typical of the pentasil structures. None of the reported zeolitic structures up to date combines these two units, please check <http://www.iza-structure.org/index.htm>. With the methodology we present in this new article, researchers will be able to prepare new hybrid materials containing combinations of zeolite units never explored before. This is what, from our point of view, makes this manuscript unique and deserving to be published in this highly ranked journal.

Below, detailed responses to each of the reviewer's comments are provided.

Comment 1. The cetyl groups (e.g. CTAB) have been widely reported to produce mesopores over the past several decades.

Response 1: While the reviewer is right about the use of cetyl groups to produce mesoporosity, to the best of our knowledge, the integration of the cetyl group and a structure directing agent in the same molecule to perform the synthesis of a mesoporous catalysts containing zeolitic units, has never been explored in the past. The most similar approach was the use of bifunctional

surfactants to direct the formation of MFI nanosheets reported by Ryoo and co-workers (*Nature* volume 461, pages 246–249 (2009)). However, our work is though completely different, a major novelty of our contribution is that this compound, in combination with interzeolite transformation, offers the possibility to prepare novel hybrid materials containing zeolitic units of different structures, which has not been reported using any other synthetic approaches.

Comment 2. On the basis of the authors' data, the formation of tetrapropyl units might be the key factor to determine the growth of MFI zeolite from FAU zeolite. However, there is no information about the formation of tetrapropyl units. Furthermore, the mesopores are dispersed in the 96 h sample. Why? Are the cetyl groups decomposed?

Response 2: As the referee correctly points out, the tetrapropyl units in the polar head of the surfactant are a key factor to perform the interzeolite transformation of FAU into MFI phase. The presence of the tetrapropyl units in the polar section (acting as SDA) of the surfactant is unequivocally revealed by its ^1H NMR spectrum (as shown in the new Figure S2). In this revised version, we have assigned and labeled all the ^1H NMR peaks, indicating which ones are due to the tetrapropyl units.

Regarding the absence of mesoporosity in the sample after the total crystallization of the MFI phase and the possible decomposition of the cetyl group pointed out by the reviewer, we have followed by TGA how the amount of CTPA^+ evolves in the samples with the time of treatment (see the new Figure S4). There are two possible causes of the reduction of CTPA^+ observed. First, the increase in the crystallinity of the sample and the consequent densification can cause the micelles dissociation. However, it is likely that individual (non-micelled) CTPA^+ molecules are still embedded into the MFI structure through their tetrapropyl units, as some organics can still be found in the final MFI material. On the other hand, it has been previously described that alkylammonium cations can undergo Hoffman degradation reactions in aqueous media at high pH values and temperatures (*Studies in Surface Science and Catalysis*, 168 (2007) 137-179) so, is also possible that some of the CTPA^+ molecules decompose during the crystallization following this mechanism.

This discussion, together with the TGA analysis (Figure S4), has been incorporated to the revised version of the manuscript.

Figure S4. TGA of samples prepared by treatment of a parent FAU zeolite with CTPABr at different times of treatment (shown in the legend).

Comment 3. In Figure S6, the FAU zeolite was amorphous after 6 h in CTAB+TPABr, and a relatively high crystallite MFI zeolite was formed after another 6 h (Figure S6), which are a little faster than those in CTPABr (39 and 48 h, Figure 1), indicating the mechanisms of CTAB+TPABr and CTPABr are similar. The differences might be caused by the different dissolved rates of FAU zeolites in CTAB+TPABr and CTPABr.

Response 3: In order to understand the differences in the kinetics of both syntheses, we have analyzed the mother liquor composition of the synthesis performed using CTAB+TPABr at different treatment times by ICP-MS. The evolution of the Si/Al ratio of these materials was calculated and compared with the one obtained for the CTPABr synthesis (see Figure S9). The presence of higher amounts of quaternary amines (as in the case of the CTAB+TPABr mixture) inhibits the desilication of the zeolite suggesting that the initial steps of both transformations should proceed through different mechanisms. When using CTPABr, the desilication found at short times of treatment suggests a solution-mediated mechanism. Contrary, the use of the CTAB+TPABr mixture do not change the Si/Al ratio of the solids so, a solid→solid transformation should occur. This trend is similar to the one observed in our previous publication (J. Am. Chem. Soc. 2022, 144, 5163–5171). These results have been included in the revised version of our manuscript.

Figure S9. Evolution of the Si/Al ratio with time for samples prepared using CTPABr (stars) or CTAB + TPABr (circles). Lines are interpolations to guide the eye.

Comment 4. Authors claim that only the use of a molecule comprised of both, the SDA group (polar head) and a mesopore forming function (aliphatic chain), that is, CTPABr, allows for interconversion of FAU into MFI and yields intermediate materials containing well-developed mesoporosity and building units from both zeolites. However, similar samples are obtained using CTAB+TPABr (Table S1 and Figure S6).

Response 4: As indicated in our manuscript (page 5, line 144), the complete transformation of the FAU zeolite into MFI cannot be achieved by using the TPABr/CTAB mixture in the synthesis, as only 70% of MFI could be obtained even after 7 days of treatment, see Figure S6D. However, as the referee correctly points out, the N₂ physisorption data shows that intermediate solids feature both well-developed mesoporosity and microporosity (even if it is in very little amount). Nevertheless, as revealed by TEM, in these materials mesoporosity and crystallinity are not in the same region. As can be observed in Figure S6E, intermediate materials prepared by this method consist of a physical mixture of both separated phases, amorphous mesoporous solid and MFI crystals.

Because this was not sufficiently clear in our manuscript, the paragraph has been changed to better explain this observation:

“To achieve the simultaneous formation of MFI and mesoporosity, the use of a TPABr/CTAB mixture seems to be a good option. However, we found that FAU cannot be completely transformed into MFI using this approach, as shown in Figure S7. Only, 70% of MFI was obtained, as evidenced by XRD and confirmed by the low development of microporosity even after 7 days of treatment (Figure S7D). In order to gain a deeper insight into the differences between both methods, the evolution of the Si/Al ratio of the

materials was compared (see Figure S8). The presence of higher amounts of quaternary amines in the CTAB/TPABr mixture inhibits the desilication of the zeolite suggesting that the initial steps of both transformations should proceed through different mechanisms. As opposed to the CTPABr case, the small change in Si/Al ratio of the solids here suggests solid → solid transformation. More importantly, as revealed by TEM, this method yielded a physical mixture of both phases, an amorphous mesoporous solid, responsible of the type IV isotherms, and separate MFI crystals (Figure S7E). Based on these observations, we concluded that only the use of a molecule comprised of both, the SDA group (polar head) and a mesopore forming function (aliphatic chain), that is, CTPABr, allows the interconversion of FAU into MFI and the formation of intermediate materials containing well-developed mesoporosity and building units from both zeolites.”

Comment 5. The data in Table S1 (12, 24, 72, and 168 h) is not in good accordance with Figure S6 (6, 12, 36, and 168 h).

Response 5: The data was carefully reviewed and Table S1 corrected.

Comment 6. A strong peak at ~26° can be observed in all the XRD patterns. There is no information about this strong peak.

Response 6: We thank the reviewer for mentioning this point. The peak at 26° corresponds to the graphite we use as internal standard in order to quantify the crystallinity of the samples. The peak has been accordingly labeled in the XRD patterns included in the manuscript and the use of graphite included in the materials characterization section.

Comment 7. In Figure S3, there are six samples. However, only five colors are identified.

Response 7: We have carefully identified all the color in the Figure S3, and the legend has been accordingly modified.

Comment 8. Figures S2, "CTEABr" should be "CTPABr".

Response 8: This typo has been corrected.

Comment 9. In page 5, line 147~150, Figure S5D and S5E should be Figure S6D and S6E, respectively.

Response 9: These typos have been corrected.

Comment 10. In Figure 2, the authors claimed that time-resolved UV-Raman spectra of the prepared solids were recorded. However, no details of the testing conditions are provided. Furthermore, these spectra are not in situ results and thereby cannot be named as time-resolved UV-Raman spectra.

Response 10: The details of the UV-Raman characterization were included in the materials characterization section; however, some details were missing, which have been included in the revised version of our manuscript.

As the referee correctly point out, our spectra are not in situ so, the time-resolved has been removed from the discussion section.

Comment 11. The letters (a, b, c, ...) in all Figures should be capitalized (A, B, C, ...).

Response 11: Following the reviewer suggestion, all the identification letters in the Figures have been capitalized.

Reviewer #3 (Remarks to the Author):

The authors reported the transformation of FAU zeolite into ZSM-5 zeolites. The transformation of one type of zeolites to another is well known in the open literature. However, the authors reported the influence of different amines on the formation of ZSM-5 and the generation of porosity to the intermediates amorphous materials. Moreover, the study showed the importance of an amorphous material in which most researchers neglect their use. Therefore, this research showed some originality. The structure of the starting material, intermediates (named as hybrid materials), and final product were first prepared and then investigated using different characterization techniques. Although the structure was fairly characterized, the manuscript required MAJOR modifications before being considered to be published:

Responses to Reviewer 3: We thank the reviewer for their helpful comments and for their positive statements about the originality of our manuscript, and the importance of the amorphous intermediates of interzeolite transformation, which has been usually neglected by researchers. We have made all the suggested changes to the manuscript and we feel they have improved both the clarity and quality of our manuscript.

Comment 1. The experimental is not well explained; if someone wants to prepare the material, I believe he/she cannot. In the "Interzeolite FAU to MFI transformations" section (p. 11, lines 343-345), the amount of each solution that constitutes the 28.5 mL solution must be stated clearly NOT only mentioning the concentration. Please do the same for other surfactants. The equation used for calculating the percent of crystallinity must be stated. Furthermore, an alternative and reliable sophisticated technique for calculating the percent of MFI and amorphous should be used instead of relying on XRD results.

Response 1: The experimental part has been rewritten in order to include all the preparation details and avoid any reproducibility problems.

Regarding the quantification of the percent of crystallinity, we have used an internal standard (graphite) to properly assess the degree of crystallinity as a percentage over a control (in our case HyZ-96, the most crystalline MFI synthesized). The use of an internal standard to quantify other phase has been widely reported in X-ray analysis, see *Powder Diffraction*, 15(3), 163-172 as an example. As mentioned in the revised version, the peak at $26^\circ 2\theta$ in the XRD patterns of all the samples corresponds to the main peak of graphite. As this was not clear in along the

manuscript, the graphite peak has been labeled in the XRD figures and the description updated in the materials characterization section, see next.

"X-ray diffraction (XRD) patterns were obtained in a powder X-ray diffractometer (Bruker AXS D8 Advance) with graphite monochromatized Cu K α radiation at 40 kV and 40 mA. A known amount of graphite (10wt%, used as an internal standard) was mixed with the samples before the analysis in order to quantify the crystallinity of the different samples (see the peak corresponding to the graphite at ca. 26° 2 θ).[1] The most crystalline MFI zeolite obtained after interzeolite transformation is completed, this is zeolite prepared by using TPABr as SDA at 96 h, was defined as 100% crystallinity and used as a reference to calculate the percentage of MFI in the intermediate samples. The formula used to calculate that percentage was: $\%MFI_{(051)} = \frac{I_{(051)x}/I_{(graphite)x}}{I_{(051)ref}/I_{(graphite)ref}} * 100.$ "

Regarding the use of an alternative technique for the determination of the amorphous phase, in this work we are interested in the amount of MFI that is obtained at different treatment times and its relationship with the textural properties of the materials so, we feel that XRD is the general technique for this purpose. We agree with the reviewer that other more sophisticated technique can be used for calculating the amount of amorphous contained in the samples; however, as we just mentioned, by using an internal standard in XRD we can quantify the amount of MFI zeolite, which is the most important feature in relation with the amount of mesoporosity. Furthermore, the volume of microporosity of the samples (a proxy of the crystallinity) also supports the data obtained by the XRD quantification.

Comment 2. The setup used for the evaluation of the catalysts was not well explained. The authors need to draw a detail schematic for the reaction setup with products analysis.

Response 2: We thank the reviewer for pointing out this missing information. The reactor and product analysis method for 1,3,5-TiPBz catalytic cracking were mentioned in our previous article (*Green Chem.*, 2020, 22, 5115-5122). Nevertheless, herein, we have drawn a schematic diagram to demonstrate internal parts of our rig intuitively. The diagram has been included in the revised version of the Supporting Information as shown in Figure S1a.

A

Figure S1. (a) Schematic representation of the catalytic system used for the catalysts evaluation.

Comment 3. "Scheme 1" in the introduction was mentioned and I do not see it in the text.

Response 3: The sentence where Scheme 1 was included has been removed from the introduction.

Comment 4. There are some typos that need to be corrected.

Response 4: A thorough revision of the manuscript has been performed in order to remove all the typos.

Comment 5. In line 105, Figure S3 is Ar physisorption NOT N2.

Response 5: The reviewer is correct, Figure S3 shows Ar physisorption isotherms. The typo in line 105 has been corrected

Comment 6. Lines 124-125, the statement is not in agreement with results represented in Figure S3; the intermediate materials showed a higher uptake at low relative pressure, say $P/P_0 = 1E-5$, in comparison with FAU ref.

Response 6: The reviewer is correct in their comment, as the intermediate materials present a higher Ar uptake at $P/P_0 < 10^{-4}$ than the FAU zeolite. That is what we mean in our previous statement 'Interestingly, the intermediates display adsorption isotherms that are between that of FAU and MFI zeolites.'. However, as it was not clear, we rephrased the sentence to better describe this result:

'We also followed the formation of hybrid zeolites by monitoring their microporosity at very low relative pressures (from $P/P_0 = 10^{-7}$) (Figure S3). FAU and MFI zeolites present very distinct adsorption profiles in the low P/P_0 range ($< 10^{-4}$), as expected from their quite different microporous architecture. Interestingly, as a result of the interzeolite transformation process, the adsorption capacity of the intermediates at low relative pressures evolves from that characteristic of a FAU zeolite to that of an MFI zeolite, confirming the evolution of the porous architecture of the solids during treatment.'

Comment 7. Figure S5 is not well labeled. It is recommended to put all figures as in Figure S5C.

Response 7: The legend of the Figure S5 (now Figure S6) has been modified to correctly identify all the samples.

Comment 8. Figure 2B is for ^{29}Si NMR, line157.

Response 8: The typo has been corrected.

Comment 9. All SEM and TEM micrographs should have the same resolution. Also, it's preferable to label the scale bar rather than mentioning that in the caption. The inter-structure of the produced materials should be confirmed by high-resolution TEM. You can find similar work in the literature <https://doi.org/10.1016/j.fuel.2018.03.161> .

Response 9: As the referee points out, in most cases, the best way to compare SEM and TEM micrographs of different samples is by observing them at the same resolution. However, it is important to highlight here that the parent FAU crystals are much smaller than the final MFI zeolite (200-500 nm for FAU vs 2-3 μm for MFI). This size dissimilarity makes very difficult to use the same magnification for all the images. In fact, the micrographs were selected to better visualize the morphological transformation of the material. Following the referee's suggestion, the scale bars have been labeled in Figure 3.

Regarding the structure of the hybrid materials, it is difficult to establish their structure by TEM analysis. Our intermediate materials do not feature the crystalline structure of both zeolites, as in the work suggested by the reviewer. The FAU crystallinity totally disappears before the MFI phase starts to appear as it is clear from our XRD analysis (see Figure 1a). For this reason, it cannot be expected to observe both crystalline structures coexisting by HR-TEM. However, as shown by UV-Raman, there are still FAU fragments in the materials when the MFI fragments start to form but either of these fragments present clear dots in SAED analysis due to lack of long-range order. Only when the MFI phase is totally formed, even in small crystals very different from the final material, it is possible to observe its SAED pattern, see next.

Figure. (left) An MFI small crystal found in the sample interconverted for 72 h and (right) the corresponding SAED pattern showing the dots corresponding to the MFI structure.

Comment 10. "These MFI building units grow over the remains of the FAU zeolite (Figure 3D)": the authors showed the dissolved fragments of the FAU zeolites over the particles, but failed to show the MFI particles? If they claim that an x % of MFI was formed, SEM and TEM micrographs should show the morphology of both mixtures and should be compared to the physical mixing.

Response 10: As explained in our previous response to the reviewer's comment, it is difficult to evaluate the material's structure by SEM and TEM (see the next image). The materials obtained at different times feature completely different morphologies. First, the totally amorphous material (HyZ-39) consists of mainly mesoporous particles with a FAU morphology. The appearance of MFI peaks in the XRD patterns occurs as a denser phase form from the previous

phase (maintaining the FAU morphology HyZ-48) which, evolve as MFI crystals, smaller in size first (HyZ-72) and larger and more agglomerated particles at the end (HyZ-96).

The physical mixtures present a totally different morphology consisting of faceted FAU crystals of small size (200-500 nm) and larger and more irregular MFI agglomerates (see the image below).

The manuscript has been revised to include these results and conclusions.

Comment 11. "As the MFI zeolite forms, the surfactant micelles are expelled out of the crystal, which causes the final material to lack mesoporosity" how does this happen?

Response 11: To answer the referee's question, we have followed by TGA how the amount of CTPA⁺ evolves in the samples with the time of treatment (see the new Figure S4). There are two possible causes of the reduction of CTPA⁺ observed. First, the increase in the crystallinity of the sample and the consequent densification can cause the micelles dissociation. However, it is likely that individual (non-micelled) CTPA⁺ molecules are still embedded into the MFI structure through their tetrapropyl units, as some organics can still be found in the final MFI material. On the other hand, it has been previously described that alkylammonium cations can undergo Hoffman degradation reactions in aqueous media at high pH values and temperatures (*Studies in Surface Science and Catalysis*, 168 (2007) 137-179) so, is also possible that some of the CTPA⁺ molecules decompose during the crystallization following this mechanism.

This discussion, together with the TGA analysis (Figure S4), has been incorporated to the revised version of the manuscript.

Comment 12. The evaluation part starting from the experimental, as mentioned earlier, still needs to be improved. The authors need to talk about the active sites responsible for the catalytic cracking of TiPBz and how these active sites differ from one sample to another. In fact, the chemical properties of all catalysts were not characterized. The porosity of a material is important to access active sites and controlling the selectivity of the products. This was clear from the reaction results over ZSM-5, which showed limited activity because the feed was unable to access the active sites. The role of acidity should be highlighted, and catalysts acidity should be determined and linked to the reaction results. The physically mixed FAU and MFI should be evaluated in the reaction and compared to the HyZ materials.

Response 12: Following the reviewer suggestion we have evaluated the performance of physical mixtures of FAU and MFI at different weight ratios, these are, FAU:MFI = 2:1, 1:1 and 1:2. The results have been incorporated into the SI (Figure S12) of our revised version. As a control, a mesoporous aluminosilicate Al-MCM-41 material with similar Si/Al ratio (ca. 40) was also tested. The conversion levels of the physical mixtures and the intermediate materials are similar. As expected, the higher the amount of FAU in the mixture, the higher the conversion obtained, owing to its larger micropore size as compared to MFI. However, the selectivities of the physical mixtures of conventional zeolites falls in the line between both zeolites indicating that, for the same conversion, lower selectivities are always obtained for the physical mixtures as compared to our catalysts, which confirms the hybrid nature of the intermediates. On the other hand, a typical amorphous mesoporous aluminosilicate (Al-MCM-41) yields similar selectivities than the intermediates but very low conversion when compared to both the zeolites and the intermediate. From these results we conclude that our materials present the best of two worlds, this is, similar conversions to highly acidic zeolites and the excellent selectivities to DiPBz which is a feature of highly accessible mesoporous materials.

Figure S12. Catalytic activity of the different materials evaluated: hybrid intermediates (39 h, light cyan; 48 h, dark cyan; 72 h, purple), FAU (black) and MFI (red) zeolites; physical mixtures of FAU:MFI at different weight ratios (1:1, brown; 2:1, green; 1:2, dark red), and an Al-MCM-41 type material (dark blue). (a) Performance of the materials in the catalytic cracking of TiPBz. (b) Evolution of the selectivity for 1,3-diisopropylbenzene with the increased conversion of TiPBz. (c) Evolution of the selectivity for cumene and benzene with the increased conversion of TiPBz.

These catalytic results indicate the presence of stronger and zeolite-like acid sites in the hybrid intermediates. To confirm this point, ^{27}Al NMR analyses were performed to study the different Al environments on the materials (Figure S13). The ^{27}Al NMR spectrum of the original FAU

sample shows the usual two bands around 60 and 0 ppm due to the presence of Al in tetrahedral (Al(IV)) and octahedral (Al(VI)) coordination environments, respectively. Noticeably, the samples prepared by interzeolite transformation do not contain octahedral Al (i.e., extra-framework species). The transformation of octahedral Al into tetrahedral Al has been widely reported for the treatment of zeolites with alkaline solutions and associated to the incorporation of extra-framework Al into the zeolite framework.[2–4] The broadening of Al(IV) signal is due to the formation of partially bonded framework Al, that is, with higher disorder., which was also found in the ^{29}Si NMR analysis (Figure 2B). Interestingly, the ^{27}Al NMR signal shifts to values more typical of the MFI structure, which indicates that the local environment of the Al sites evolves towards that found in an MFI structure. This confirms the Raman spectroscopy results (Figure 2A), even at short times (39h), when XRD peaks are not observed. Finally, Al-MCM-41 material contains ca. 50% of octahedral aluminum (Al(VI)), where even our XRD amorphous material (HyZ-39) only contains tetrahedral Al(IV). The exclusive presence of only Al(IV) in our intermediate materials can be related to their superior catalytic performance, as suggested by the much higher conversion of HyZ-39 vs Al-MCM-41 (60% vs 10% after ten pulses, Figure S12) both showing similar porosity and lacking XRD crystallinity. These results have been included in the revised version of the manuscript.

Figure S13. ^{27}Al NMR spectra of the parent FAU, the intermediate samples treated at different times and the final MFI zeolite. The spectrum of an amorphous mesoporous Al-MCM-41 material has been included for comparison purposes.

References

- [1] R.S. Winburn, D.G. Grier, G.J. McCarthy, R.B. Peterson, Rietveld quantitative X-ray diffraction analysis of NIST fly ash standard reference materials, *Powder Diffr.* 15 (2000) 163–172. <https://doi.org/DOI: 10.1017/S0885715600011015>.

- [2] Z. Zhang, X. Liu, Y. Xu, R. Xu, Realumination of dealuminated zeolites Y, *Zeolites*. 11 (1991) 232–238. [https://doi.org/10.1016/S0144-2449\(05\)80224-6](https://doi.org/10.1016/S0144-2449(05)80224-6).
- [3] V. Calsavara, E.F. Sousa-Aguiar, F.M. Fernandes, Reactivity of USY extraframework alumina in alkaline medium, *Zeolites*. 17 (1996) 340–345. [https://doi.org/10.1016/0144-2449\(96\)00071-1](https://doi.org/10.1016/0144-2449(96)00071-1).
- [4] S.M.C. Menezes, V.L. Camorim, Y.L. Lam, R.A.S. San Gil, A. Bailly, J.P. Amoureux, Characterization of extra-framework species of steamed and acid washed faujasite by MQMAS NMR and IR measurements, *Appl. Catal. A Gen.* 207 (2001) 367–377. [https://doi.org/10.1016/S0926-860X\(00\)00676-1](https://doi.org/10.1016/S0926-860X(00)00676-1).

REVIEWERS' COMMENTS

Reviewer #1 (Remarks to the Author):

I was satisfied with the revision. It can be accepted.

Reviewer #2 (Remarks to the Author):

The revised manuscript has been improved greatly. I recommend publishing this manuscript after minor revision.

The different mechanisms between CTPABr and CTAB+TPABr should be further identified. Authors claimed that only 70% of MFI was obtained after 7 days (0.4 g of CTAB and 0.222 g of TPABr). However, the effect of the amounts and ratios of CTAB+TPABr is not investigated. Possibly, a larger or lower amount of CTAB+TPABr can lead FAU completely transform into MFI.

Reviewer #3 (Remarks to the Author):

Thanks to the authors for their hard work. They have addressed most of the comments.

REVIEWER COMMENTS

Reviewer #2 (Remarks to the Author):

The revised manuscript has been improved greatly. I recommend publishing this manuscript after minor revision.

Response to Reviewer 2: We wish to thank the reviewer for recommending the publication of our manuscript.

Comment. The different mechanisms between CTPABr and CTAB+TPABr should be further identified. Authors claimed that only 70% of MFI was obtained after 7 days (0.4 g of CTAB and 0.222 g of TPABr). However, the effect of the amounts and ratios of CTAB+TPABr is not investigated. Possibly, a larger or lower amount of CTAB+TPABr can lead FAU completely transform into MFI.

Response: In this study, we calculated the amounts of CTAB and TPABr to obtain the same molar concentration as in our CTPABr experiment, for a fair comparison. However, as suggested by the reviewer, further study of the CTAB+TPABr ratios may lead to the complete transformation of FAU into MFI. However, previous research has shown that mixing conventional cationic surfactants (CTAB) with organic structure agents (TPAOH or TEAOH) in zeolite synthesis often leads to phase separation of amorphous mesoporous material and crystalline microporous zeolite (see J. Mater. Chem., 2006, 16, 2235), which is what we observed in our study (Supplementary Figure 6E).

We agree with the reviewer that the study of the different mechanisms between CTPABr and CTAB+TPABr is an interesting research topic, but it is a long study outside the scope of this work. To avoid phase separation, the synthesis conditions must be carefully evaluated and the use of co-solvents and co-surfactants assessed. We have modified the discussion about the use of CTAB + TPABr to clarify this point.

“Finally, while the use of a TPABr/CTAB mixture may appear promising for simultaneous formation of MFI and mesoporosity, our results indicate that it is not sufficient for complete transformation of FAU into MFI. As demonstrated by XRD analysis and low development of microporosity after 7 days of treatment (Supplementary Figure 7D), only 70% of MFI was obtained. To further understand the differences between the two methods, we compared the evolution of the Si/Al ratio of the materials (Supplementary Figure 8). The high levels of quaternary amines in the CTAB/TPABr mixture were found to inhibit the desilication of the zeolite, indicating that the initial stages of both transformations likely proceed through different mechanisms. Unlike the CTPABr case, the small change in the Si/Al ratio of the solid suggests a solid-to-solid transformation. Furthermore, TEM analysis revealed that this method produced a physical mixture of both phases: an amorphous mesoporous solid (responsible for type IV isotherms) and

separate MFI crystals (Supplementary Figure 7E). Previous research has shown that mixing conventional cationic surfactants (CTAB) with organic structure directing agents in zeolite synthesis often leads to phase separation of amorphous mesoporous material and crystalline microporous zeolite,^{21,22} which is consistent with our findings. Based on these observations, we conclude that only the use of a molecule that includes both a polar head (SDA group) and a mesopore-forming function (aliphatic chain) - CTPABr - allows for the interconversion of FAU into MFI and the formation of intermediate materials with well-developed mesoporosity and building units from both zeolites under equivalent synthetic conditions."